# Reproducibility report of *From goals, waypoints & paths to longterm human trajectory forecasting* for ML Reproducibility Challenge 2021

**Scope of Reproducibility**

The following paper is a reproducibility report for From Goals, Waypoints & Paths To Long Term Human Trajectory Forecasting [9]. The basic code was made available by the author at *this https url*. We have verified all claims and results from the experiments mentioned in the paper to support the claims. The central claim of YNet is that it sets state-of-the-art short and long-term prediction standards by a multi-modal network employing both segmentation matrices and past trajectory heat-maps together.

**Methodology**

The model essentially combines a segmentation map and past trajectory heatmaps to encode a combined input to three sub-networks modelled after the U-Net architecture [12]. The author's code was used to benchmark the claims, and some experiments were performed thereafter. Free-to-use platforms like Google Colaboratory and Kaggle were used to train these models. We have reproduced the code base in PyTorch Lightning (originally in PyTorch Ignite) and found consistent results across the board.

**Results**

Through our testing, we were able to come within 2% of the proposed metrics on certain datasets like Stanford Drone Dataset (SDD) and ETH/UCY, implying the author's claims are sanguine and reproducible on varied hardware. However, certain results such as long-term predictions on the Intersection Drone (InD) dataset were quite different; the probable reasons for which have been discussed.

**What was easy**

Obtaining the proposed results on the SDD and InD datasets was easy. Well-documented interactive notebooks for training and testing along with the requisite data for SDD were provided with the codebase. The code could be run with minimal changes overall.

**What was difficult**

The codebase and data provided by the authors were incomplete, and contained various redundancies and unused methods, making it difficult to follow. The requisite code and data required to reproduce the experiments on the ETH and UCY datasets were completely missing. These factors were compounded by stringent computational power requirements, which were difficult to fulfill for students without access to server-grade computation.

**Communication with original authors**

Attempts were made to contact the authors regarding some doubts, which went without response. Running the experiments thereafter were done based on our understanding of the paper and code.

---

# 1   Introduction

The paper reproduced in this report aims to tackle multiple pedestrian trajectory predictions using rich multi-modal predictions for the use of autonomous vehicles, social robots, etc. Earlier approaches to this problem have been auto-regressive in nature [1][8][14], i.e., using n points (or, analogically, data from the last t seconds) from the dataset to produce the next immediate point, and recurring this process.

In this paper, the trajectory distribution, viz. the path taken by a pedestrian is conceived to have been influenced majorly by two factors:

- Epistemic: The conscious will of the pedestrian to reach a particular goal.

- Aleatoric: The unknown and unexpected changes in the environment influencing the path they take to reach the goal.

The proposed architecture incorporates this multi-modality. An explicit probability distribution of the many possible broad future trajectories is predicted first (modelling the *where* of the agent). Then, random future points of the trajectory are taken in conjunction with the sampled way-points to obtain probability maps over the remaining predicted points (modelling the *how* of the agent).

To formulate this report, we have experimented on the author's code by adding/removing social pooling layers and employing visualisation tools. We have tried the unique idea of multi-dataset training wherein we train the model for long on a particular dataset, and then immediately introduce it to a completely new dataset. We also performed some experiments such as shifting the prediction origin to different previously predicted points instead of the one closest in time to the present. These experiments are explained in detail in the following sections.

# 2   Scope of reproducibility

The problem of multi-modal trajectory prediction is key to unlocking vehicular intelligence and autonomous navigation. The problem this particular paper aims to address is finding pedestrian trajectories in an environment crowded with similar and/ or different interacting agents. By extension, the scope of using such architecture is beyond pedestrians, as virtually any human or non-human agent navigating crowded terrains that may benefit from segmentation may employ such mechanics of trajectory prediction.

The central claims of the paper can be summarized as follows:

- Conditioned way-point predictions: The model performs better than previous works as trajectories are conditioned in a two-stage manner, with aleatoric predictions conditioned upon epistemic ones. This provides stricter constraints on the final set of predictions by modelling them explicitly, as opposed to SGAN [5], SoPhie [13] and other attention based mechanisms that produce a diverse set of trajectories.

- Scene Segmentation: The model performs better than contemporary models as semantic information about the scene is accounted for. The paper considers as input, both the segmentation map and trajectory heatmap of probabilities. The segmentation step is a novel addition that classifies the possible trajectory avenues of the pedestrian. Intuitively, this can be thought of as follows: Given a predicted valid goal of the pedestrian, he is highly unlikely to climb a wall to achieve it. Rather, he shall traverse his current course (say, a park track). The segmentation steps performs better than previous non-segmented attention mechanisms.

- Long Term Prediction: The model uses these techniques to achieve significantly improved results on long horizon trajectory prediction as well as short horizon.

# 3   Methodology

We used the GitHub repository provided by the author as the base. However, it only contained the base model for results on the different data sets. In order to reproduce the rest of the experiments, we had to make changes accordingly.

## 3.1 Model descriptions

The problem of multi-modal trajectory prediction can be formulated as prediction of future trajectory given past positions of pedestrians in the scene. This section has been referenced from the original paper [9] (Section 3).

The scene image is first processed by a segmentation network, producing segmentation map S (dividing the image in various classes) of the same spatial size as image. In a parallel branch the past trajectories are embedded in a trajectory heatmap. Concatenation of both produces the heatmap tensor $H_s$. For each frame $n$ in the input, the heatmap is calculated as

$$\mathbf{H}(n, i, j) = 2 \frac{||(i,j) - \mathbf{u}_n||}{max_{(x,y)\epsilon\mathcal{I}}||(x,y) - \mathbf{u}_n||} \tag{1}$$

The heatmap and semantic maps are concatenated and fed into the encoder branch $U_e$ of the network.

The subsequent architecture consists of 3 sub-networks $U_e$, $U_g$ and $U_t$. $U_e$ which is an encoder like U-Net [12] is used in the model architecture to process the tensor $H_s$. It has a total of $N_{U_e}$ blocks, it reduces the dimensions of the H × W to $H_U \times W_U$ and increases the channel depth. The final representation is termed as $H_{U_e}$ which is then passed in the goal decoder $U_g$ and the trajectory decoder $U_t$.

The next step is termed as the "Goal & Waypoint Heatmap Decoder" in which the output maps of $U_e$ at various spatial resolutions are passed in $U_g$ which is modelled from U-Net. The output is passed through a deconvolution layer, which involves the application of a transpose convolution, effectively expanding the previous feature map, spatially doubling the resolution in every block. The encoder map from the respectively sized input layer is concatenated. To attain the final resolution of the goal, heatmap feature merging is done. Therefore, it can be said a U-Net block consists of deconvolution, feature merging and convolution layers.

The final branch $U_t$ is termed as the "Trajectory Heatmap Decoder". The waypoint distributions from $U_g$ are sampled using the softargmax operation

$$\texttt{softargmax}(X) = \left( \sum_i i \frac{\sum_j e^{X_{ij}}}{\sum_{i,j} e^{X_{ij}}}, \sum_j j \frac{\sum_i e^{X_{ij}}}{\sum_{i,j} e^{X_{ij}}} \right)$$

A heatmap tensor $\mathbf{H}_{U_g}$ is generated using these samples. Each heatmap is downsampled to its corresponding size from the architecture. These heatmaps are concatenated with the respective blocks from $\mathbf{H}_{U_e}$ which goes through $U_t$ for a decoding phase.

## 3.2 Datasets

All annotations were preprocessed to match the format ['trackId', 'frame', 'x', 'y', 'sceneId', 'metaId']. For experiments based on varying the prediction window, the data was preprocessed with different trajectory lengths.

**Stanford Drone Dataset (SDD)**: [11] The dataset by default contains annotations for 10,300 unique agents across 6 classes, of which 5232 belong to the class of pedestrians. Trajectories are sampled at FPS = 30 in 2D image coordinates. We use the pre-processed data provided by the authors, which has been downsampled to FPS = 2.5 for short term training and FPS = 1 for long term. The lengths of the input sequences $n_p$ are 8 and 5 respectively, while the those of the output $n_f$ are 12 and 60 respectively. All trajectories not belonging to the pedestrian class or of insufficient length ($< n_f + n_p$) are dropped. The midpoints of the bounding boxes are considered to be the ground truth positions. Trajectories are split at temporal discontinuities and a staggered sliding window is used to split long trajectories. The resultant is a set of 1502 trajectories. A semantic map with 5 classes is generated. There is a train/test split of 30 scenes for training and 17 for testing.

**Intersection Drone Dataset (InD)**: [3] The dataset by default contains 11,500 trajectories across 3 classes, in 32 scenes at 4 distinct locations. Trajectories are sampled at FPS = 25 in 2D world coordinates. We perform the preprocessing described in the paper, which involves downsampling the data to FPS = 1 for $n_p = 5$ and $n_f = 30$, filtering out non-pedestrians, filtering out short ($< n_f + n_p$) trajectories, splitting long (using the sliding window technique) and discontinuous trajectories. The coordinates are brought into image coordinates by using the scale factors and cropping

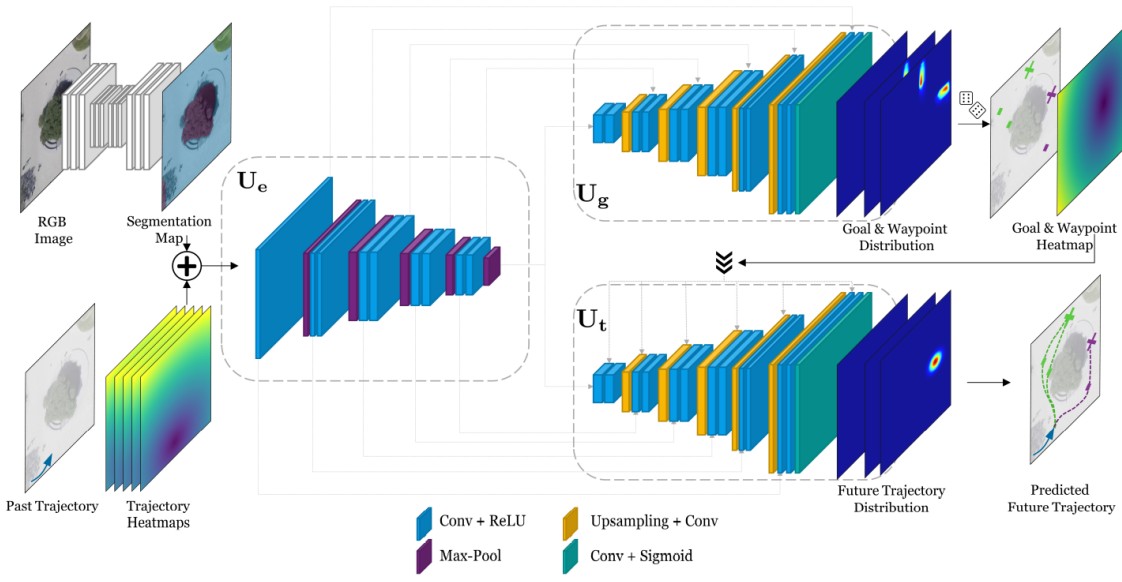

Figure 1: Graphical view of model architecture (referenced from parent paper)

parameters provided in the paper as cited. The resultant is a set of 1396 trajectories. The scenes of one location (ID 4) are used for testing while those of the remaining 3 are used for training.

**ETH and UCY datasets (ETH/UCY)**: Combined, the dataset contains trajectories for 1536 pedestrians, in 9 scenes at 5 distinct locations. Trajectories are sampled at FPS = 2.5 in 2D world coordinates. The authors claim to use preprocessed data from [5], however this is not usable on account of being normalized with unknown parameters and being in the incorrect format. We instead use preprocessed data provided by the authors in response to an issue raised on the GitHub repository. We take $n_p = 8$ and $n_f = 12$ as per the paper. We use the homography matrices provided with the datasets to transform the pixels into world coordinates. A leave-one-out cross-validation strategy is employed.

### 3.3 Hyperparameters

Several hyperparameters were experimented with in this paper. Those of particular importance are the following:

- $K_e$: The model aims to produce $K_e$ possible future trajectories due to the epistemic uncertainty of final goal. This is a hyperparameter that can be tuned to find the optimal value for any given probability map as input.

- $K_a$: After the end-point prediction distribution, the path(s) taken to reach the most likely of them constitutes absolute randomness when not conditioned on aleatory factors and environmental interactions. Thus, given the goal, the model produces $K_a$ separate predictions for path.

- $T$: The temperature parameter T during sampling can be visualised as a scaling factor for the generated heat-map. It is used to control the trade-off between diversity and precision; a lower value meaning the predictions are condensed in a smaller spatial density and vice versa. Intuitively, a higher value of T should be used for long-term predictions, but this parameter can still be tuned to gauge the power of the model.

All hyper parameters were tuned by random searches and heuristic guesses instead of brute/ grid searches or Bayesian techniques, mainly due to constraints posed by very high computational resource requirements. However, the trends in accuracy could still be predicted and reasoned as the effects of changing the values were both experimentally visible and logically explainable. These points have further been discussed in the Results and Discussions sections.

| Hyperparameters | Value |
|---|---|
| $K_e$ | 5 or 20 |
| $K_a$ | 1 |
| $T$ | 1.8 |
| Number of epochs | 100 |
| Batch Size | 8 (4 on long term) |
| Learning Rate | 1E-4 (optimal) |
| Semantic Classes | 6 (3 for ETH/UCY) |
| Waypoints | 11 |

Table 1: Hyperparameters used in the paper

## 3.4 Experimental setup and code

The code for this experiment is set-up mainly in the `train.py`, `evaluate.py` and `test.py` Python files that import helper methods defined in python files in the utils folder. Python notebooks are given to facilitate running different parts of the code. Detailed instructions about each, the presence of pre-trained weights and/ or pre-processed files and other relevant information is given in the ReadMe section of the repository.

The main metrics of interest are the ADE (mean L2-norm distance between all future ground truth and predicted points) and FDE (mean L2-norm distance between final future ground truth and predicted points). The accuracy of all experiments are validated with these metrics, where a lower value means a more accurate result.

## 3.5 Computational requirements

All experiments were run using Google Colaboratory, whose back-end has the Tesla P100 GPU. The technical specification of the GPU is that it has 3584 CUDA cores, 16GB CRAM and a 4096-bit memory interface.

The run-times we faced on such a setup for the different experiments is quite long. For example, running the code without any ablations on the SDD dataset took roughly 10 minutes for a single epoch. Of course, this value is dependant on other hyperparameters such as batch size.

# 4 Results

The results we obtained are listed below. Upon comparison, we are confident that the results resemble those in the paper. However, due to the lack of extensive computational capabilities, we were forced to limit our training to a fraction of what was done in the original paper. Despite this, we have deeply analysed fitting and convergence trends and are confident that the model does at least as well as claimed, and even better in some experiments.

All results were logged with ease with the WandB solution [2]. In general, extensive overview of error trends could be gauged from auto-generated graphs, which cemented our beliefs of convergence and correctness.

## 4.1 Results reproducing original paper

### 4.1.1 Performance of model as compared to baselines

Our reproduced model functions better than all previous baselines, and satisfactorily close to the reults of YNet as cited in the paper.

| | K=20 | | | | | K=5 | | | |
|---|---|---|---|---|---|---|---|---|---|
| | P2TIRL[4] | SimAug[7] | PECNet[10] | Y-net (Paper's) | Ours' | TNT[15] | PECNet[10] | Y-net (Paper's) | Ours' |
| ADE | 12.58 | 10.27 | 9.96 | 7.85 | **8.85** | 12.23 | 12.79 | 11.49 | **12.36** |
| FDE | 22.07 | 19.71 | 15.88 | 11.85 | **12.23** | 21.16 | 29.58 | 20.23 | **20.18** |

Table 2: Short temporal horizon forecasting results on SDD

### 4.1.2 Performance of model for different datasets of ETH/UCY: Importance of social masking

This table is produced separately because it addresses the importance of social masking. The paper results are with masking, while ours are without.

| | ADE | | FDE | |
|---|---|---|---|---|
| | YNet (Paper's) | Our's | YNet (Paper's) | Our's |
| ETH | 0.28 | **0.38** | 0.33 | **0.60** |
| HOTEL | 0.10 | **0.69** | 0.14 | **0.97** |
| UNIV | 0.24 | **0.35** | 0.41 | **0.61** |
| ZARA1 | 0.17 | **0.31** | 0.27 | **0.423** |
| ZARA2 | 0.13 | **0.34** | 0.22 | **0.57** |

Table 3: Short temporal horizon forecasting results on several datasets of ETH/UCY without social masking.

### 4.1.3 Turning TTST and CWS on and off

TTST and CWS are heuristics designed to improve sampling. TTST encourages clustering samples in high-density regions by roughly thresholding and clustering the probability distribution. CWS discourages sampling erratic trajectories by assuming points to be sampled between two known points roughly divide the line segment joining them. The roughness is modelled using Gaussian distributions. We experiment with various possible state to verify their effect on error.

| | SDD (paper's / ours) | | | IND (paper's / ours) | |
|---|---|---|---|---|---|
| TTST | x | x | ✓ | x | ✓ |
| CWS | x | ✓ | ✓ | ✓ | ✓ |
| ADE | 65.00 / **46.52** | 52.31 / **46.67** | 47.94 / **46.52** | 17.77 / **4.85** | 14.99 / **4.90** |
| FDE | 86.98 / **62.23** | 86.98 / **63.23** | 66.71 / **62.23** | 28.52 / **8.85** | 21.13 / **9.23** |

Table 4: Effect of TTST and CWS on SDD on inD

### 4.1.4 Hyperparameter tuning - $K_a$, $K_e$ and $T$

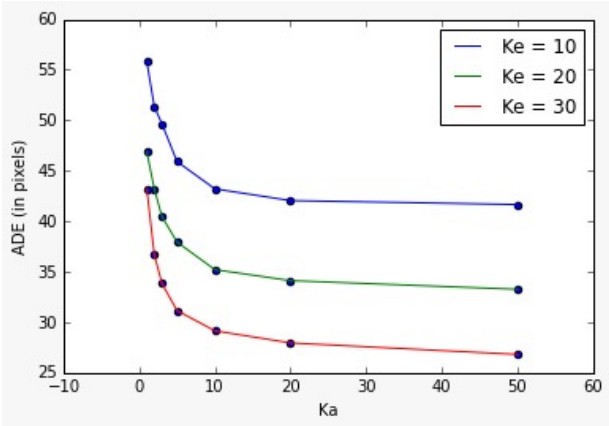

Figure 2: Variation of ADE with grid search wise changes in $K_a$, $K_e$

## 4.2 Results beyond original paper

### 4.2.1 Generalization

One of our main findings beyond the paper was the generalizing power of the model, abstracted by its potential to be used as a transfer learning model. Given that the data can be processed in a similar manner outside the domain of the

actual model, we observed much-improved results when trained for a very short time on a completely new dataset. To explore this further the idea of Fine-tuning was explored in which once the Y-net model was trained on Dataset A, the final weights were considered as the pretrained weights for a new training and the model was further trained on Dataset B for very few epochs.

In this way the model not only remembered the previous training features but also adapted the conditions for the new dataset. This method proved to improve the performance of the model and is computationally very inexpensive.

| Model Trained on | Epochs | Model tested on | No. of Epochs further trained for | ADE | FDE |
|---|---|---|---|---|---|
| inD | 300 | inD | 0 | 14.99 | 21.13 |
| SDD longterm | 300 | inD | 0 | 10.67 | 17.21 |
| SDD longterm | 300 | inD | 1 | 5.032 | 8.767 |
| SDD longterm | 300 | inD | 2 | 4.967 | 8.844 |
| SDD longterm | 300 | inD | 3 | 4.822 | 8.699 |
| SDD longterm | 300 | inD | **4** | **4.59** | **8.090** |

Table 5: Long term trajectory forecasting Results on transferred dataset

# 5 Visualisations

Some real world trajectories on actual reference images are provided below. The lines are enlarged for clarity. It can clearly be seen that the model works fantastically well in real-life scenarios to predict trajectories. Segmentation has worked well in these cases, with no class overlap except cases when the trajectory itself goes across two different semantic classes like in figure (b).

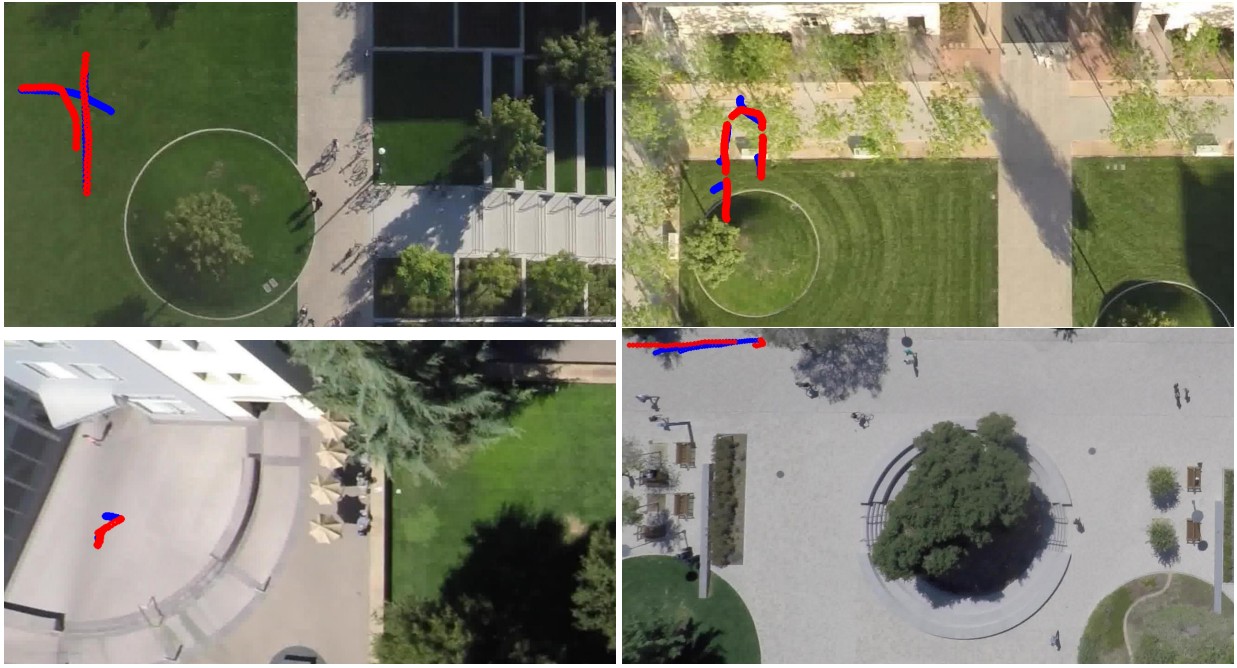

Figure 3: SDD ground truth (red) and predictions (blue) on (a) hyang-0-1 (b) hyang-1-1 (c) gates-2-2 (d) coupa-1-3 reference images. Figure (c) is a short-term prediction with minimal movement, while the other three are long-term with temporal horizon of 30 seconds.

# 6 Discussion

Many obstacles were faced in the reproduction of the results, particularly for students with limited access to server grade computational resources. The codebase and data had many redundancies and omissions, requiring some experiments to

be recreated from scratch. Despite these challenges, our experiments achieve a satisfactory reproduction of the paper. However, some discrepancies were observed. Our results on the InD dataset were significantly better than those cited in the paper. This may be due to model optimizaitons in PyTorch Lightning or fortunate random initialization of weights.

The experiments on the SDD and ETH/UCY datasets were found to be consistent with the paper. The predicted trajectories represented state-of-the-art accuracy, even more so with the TTST and CWS sampling techniques enabled. We observed enhanced accuracy with dataset dilution and marginal training on a new dataset. The long-term prediction results were viable, viz. significantly better than contemporary models, enabling this model to be used in a real-time prediction stack for trajectory prediction.

## 6.1 Further discussion on the results

Table 4 confirms that the usage of TTST and CWS markedly improves the accuracy of the final results by increasing the tendency to draw samples from relevant points in the probability distribution. However, this comes at the cost of increased computational complexity.

The model is robust towards changes in context and hence can be extended to a wide variety of applications, as evidenced by Table 5. Good transfer performance indicates the architecture is the dominant factor in our results as opposed to extraneous factors such as sampling techniques, demonstrating its strength. This experiment also highlights the potential of transfer learning in improving neural network performance, especially for complex models like this, which reap the benefit of carrying over a dense field of features.

The crux of the paper, the predictive power of the chosen multi-modality, is succinctly demonstrated by Figure 2. We see a marked improvement in inference as we increase both $K_a$ and $K_e$ independently of each other. This direct relationship indicates that the approach of sequentially predicting epistemic and aleatoric distributions is significant, and verifies this paper's contribution to the state-of-the-art of pedestrian trajectory prediction.

There are significant increases in all errors upon removing social masking and pooling as seen in table 3. This is a central claim of a paper, i.e. aleatory interactions from the surroundings are a crucial factor in determining the best path taken. Modelling them using the segmentation ResNet-101 [6] is evidently better than using deterministic criteria to model these interactions.

## 6.2 What was easy

The experiments on the SDD and InD datasets required minimal effort to reproduce. The authors provide interactive Python notebooks for training and testing, along with all the requisite scripts and most of the data to run them. Due to the modularity of the code, performing the ablation study was also easy.

## 6.3 What was difficult

There were significant challenges faced in this reproduction. Due to limited computational resources, training the extensive CNN was problematic (mainly owing to a per-epoch training time of 30-60 minutes, despite a small batch size of 4). Further, the code, data, pre-trained weights, semantic maps, semantic models for the experiments on the ETH/UCY datasets were missing from the repository, rendering it impossible to exactly reproduce the authors' experiments. The paper suggests preprocessed data from [5] be used, however we found it was unusable as that data had been normalized with unknown parameters. The preprocessing functions provided by the authors for ETH/UCY could not be used as it was not possible to fulfill some arguments. There was an error in the pre-trained weights provided for the long term SDD experiment, which caused a tensor dimension mismatch during testing.
The codebase contained some unused methods. There were some redundant parameters, for example `batch_size` $= 4$ in the yaml configuration file and `BATCH_SIZE` $= 8$ declared in the training notebooks. Some lines of code were commented-out without documentation. These factors made it difficult to follow and debug the code.

## 6.4 Communication with original authors

Multiple attempts to contact the authors were made over a 2 month period. Some doubts with the paper and the absence of ETH/UCY experiments from the codebase were raised. However, there was no response from the authors.

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
