# OpenReview forum: "From Goals, Waypoints & Paths To Long Term Human Trajectory Forecasting"
_ML_Reproducibility_Challenge/2021/Fall — RC2021_

### Official Review · Reviewer_sikw · 2022-03-01
**reasonable reproduction results; largely confirms the original paper**

**Rating:** 6
**Confidence:** 4

**Review:**

1. Reproduction results:
The paper does a reasonable job reproducing the results: Table 2 is within 12%, Table 4 is even better than the original paper. Specifically, the reproduced results on the InD dataset were significantly better than the original paper. The authors speculated that this may be due to model optimizations in PyTorch Lightning or fortunate random initialization of weights. It is a bit disappointing that authors did not dig deeper and validate their hypothesis.

For Table 3, the authors need to explain why showing the original paper's results with social masking while presenting theirs without.

2. Reproduction challenges:
The authors mentioned that the code, data, pre-trained weights, semantic maps, semantic models for the experiments on the ETH/UCY datasets were missing from the repository, rendering it impossible to exactly reproduce the authors’ experiments. They also found that preprocessed data from [5] was unusable as that data had been normalized with unknown parameters. The preprocessing functions provided by the authors for ETH/UCY could not be used as it was not possible to fulfill some arguments. There was an error in the pre-trained weights provided for the long term SDD experiment, which caused a tensor dimension mismatch during testing.

3. Clarity:
The paper is mostly readable. However, there are some gaps, e.g. notations in Eqn. 1 are not explained.

4. Quality, originality and significance of the work:
The authors made a good effort and were able to reproduce main results. This is a good contribution. Further, the paper also
discover the generalizing power of the model and the model has the potential to be used as a transfer learning model.

---

### Meta-Review · Area_Chair_Qzur · 2022-04-09

**Recommendation:** Accept
**Confidence:** 3

**Metareview:**

This report does a good job reproducing the original work, and has extensive experiments which, interestingly, at times even exceed the performance found in the original paper.

---

### Decision · Program_Chairs · 2022-04-09

**Decision:**

Accept

**Comment:**

Following the recommendation of reviewers and meta-reviewer, the paper is accepted for ML Reproducibility Challenge 2021, and will be published in the upcoming special edition of ReScience Journal.